# On the velocity at wind turbine and propeller actuator discs

Gijs A.M. van Kuik

Duwind, Delft University of Technology, Kluyverweg 1, 2629HS Delft, NL

**Correspondence:** Gijs van Kuik (g.a.m.vankuik@tudelft.nl)

**Abstract.** The first version of the actuator disc momentum theory is more than 100 years old. The extension towards very low rotational speeds with high torque for discs with a constant circulation, became available only recently. This theory gives the performance data like the power coefficient and average velocity at the disc. Potential flow calculations have added flow properties like the distribution of this velocity. The present paper addresses the comparison of actuator discs representing propellers and wind turbines, with emphasis on the velocity at the disc. At a low rotational speed, propeller discs have an expanding wake while still energy is put into the wake. The high angular momentum of the wake, due to the high torque, creates a pressure deficit which is supplemented by the pressure added by the disc thrust. This results in a positive energy balance while the wake axial velocity has lowered. In the propeller and wind turbine flow regime the velocity at the disc is 0 for a certain minimum but non-zero rotational speed .

At the disc, the distribution of the axial velocity component is non-uniform in all actuator disc flows. However, the distribution of the velocity in the plane containing the axis, the meridian plane, is practically uniform (deviation $< 0.2$ %) for wind turbine disc flows with tip speed ratio $\lambda > 5$ , almost uniform (deviation $\approx 2$ %) for wind turbine disc flows with $\lambda = 1$ and propeller flows with advance ratio $J = \pi$, and non-uniform (deviation $5$ %) for the propeller disc flow with wake expansion at $J = 2\pi$. These differences in uniformity are caused by the different strengths of the singularity in the wake boundary vorticity strength at its leading edge.

## 1   Introduction

The start of rotor aerodynamics dates back more than 100 years, when the concept of the actuator disc to represent the action of a propeller was formulated by Froude (1889). In this concept the disc carries only thrust, no torque. Based on this Joukowsky (1918) published the first performance prediction that still holds today, for a hovering helicopter rotor or a propeller without forward speed. Two years later Joukowsky (1920) and Betz (1920) published the optimal performance of discs representing wind turbines, for which reason it is called the Betz-Joukowsky maximum (Okulov and van Kuik (2012)). The names of Betz and Joukowsky are also connected with the two concepts for actuator discs with thrust ánd torque. The model of Betz (1919) was similar to the vortex model of Prandt for an elliptically loaded wing. This gives an induced velocity which is constant over the wing span, resulting in minimum induced drag. In Betz's model each rotor blade is represented by a lifting line such that the vortex sheet released by the blade has a constant axial velocity. Joukowsky (1912) developed the vortex model of a propeller based on a horseshoe vortex of a wing. In his model each blade is modelled by a lifting line with constant circulation.

The constant circulation model of Joukowsky as well as the constant velocity model of Betz represented the ideal rotor. It was not yet possible to compare the models and to conclude which was was best. Both models were valid only for lightly loaded rotors as wake expansion or contraction was neglected. A solution for the wake of Betz's rotor , still restricted to lightly

loaded propellers, was presented by Goldstein (1929). The non-linear solution, so including wake deformation, was published by Okulov (2014) and Wood (2015). A comparison of the models of Betz and Joukowsky for rotors was presented by Okulov et al. (2015, chapter 4) showing that Joukowsky rotors perform somewhat better than Betz rotors when both operate at the same tip speed ratio. The same conclusion was drawn for actuator discs by van Kuik (2017): at low tip speed ratio the Joukowsky disc performs somewhat better than the Betz disc. For increasing tip speed ratios, both models become the same as they converge

to Froude's actuator disc.

The Joukowsky and Froude discs still are subjects for research as many modern design and performance prediction codes are based on it, see e.g. Sørensen (2015). Over the last decades the disc received most attention from the wind energy research community, but recently new propeller research on the actuator disc concept has been published, see Bontempo and Manna (2018a, b, 2019). The performance aspects are known by many studies using momentum theory, vorticity or CFD methods.

Experimental verification is shown by e.g. Lignarolo et al. (2016) and Ranjbar et al. (2019). Recent research aims for deriving efficient tip corrections, see e.g. Moens and Chatelain (2018), Zhong et al. (2019), or for configurations including a hub, Bontempo and Manna (2016), or duct, Dighe et al. (2019).

The present paper addresses the topic which received the least attention: the velocity distribution at the disc. The paper is part of a sequence of papers, starting with van Kuik and Lignarolo (2016) concerning flows through wind turbine Froude

discs calculated by a potential flow method, followed by van Kuik (2017) concerning the momentum theory and potential flow calculations for wind turbine Joukowsky discs, and the conference paper van Kuik (2018a) where the extension to propeller discs was presented. The latter paper was not yet conclusive in the explanation of the difference between wind turbine and propeller discs regarding the velocity distribution at the disc: for wind turbine discs the velocity vector in the plane containing the disc axis, the meridian plane, seems to be uniform, while it seems non-uniform for propeller discs. Compared to van

Kuik (2018a) all calculations have been redone at equal, highest possible, accuracy, leading to slightly different quantitative conclusions and a consistent explanation for the (non-)uniformity of the velocities at the actuator discs. Some of the content of van Kuik (2018a) regarding the average velocity at the disc is repeated, in order to make the paper readable independent of the previous papers. The open-access book van Kuik (2018b) contains the content of all papers mentioned in this paragraph.

Section 2 presents the equations of motion and the coordinate system. Section 3 discusses the average velocity at the disc,

from infinitely high to very low rotational speeds, followed by section 4 treating the velocity distribution for both actuator disc modes. Section 5 analyses the differences observed between wind turbine and propeller discs, followed by the concluding section 6.

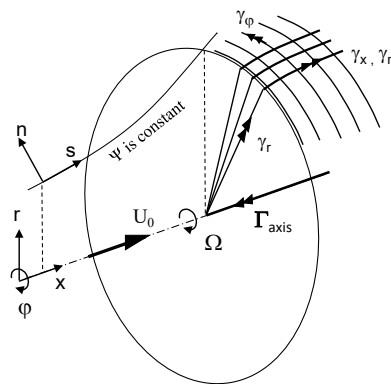

**Figure 1.** The coordinate system of an actuator disc acting extracting energy. $\Psi$ is the Stokes stream function. All vectors are in positive direction except $\Gamma_{axis}$ and $\gamma_\varphi$.

## 2 Equations of motion

Figure 1 shows the coordinate systems. The disc is placed perpendicular to the undisturbed velocity $U_0$, rotating with angular velocity $\Omega$. All vectors are in positive direction, apart from $\Gamma_{axis}$, the vortex at the axis, and $\gamma_\varphi$, the azimuthal component of the wake boundary vortex sheet. The steady Euler equation is valid:

$$\rho(\boldsymbol{v}\cdot\boldsymbol{\nabla})\boldsymbol{v} = -\boldsymbol{\nabla}p + \boldsymbol{f}, \tag{1}$$

with $\boldsymbol{f}$ the force density, in this case distributed at the disc with thickness $\epsilon$. The velocity is presented in the cylindrical coordinate system with $x$ pointing downstream: $\boldsymbol{v} = \{v_x, v_r, v_\varphi\}$. $\rho$ is the flow density, $p$ the pressure. In some of the equations dimensionless variables for the axial velocity will be used: $u_d = v_{x,d}/U_0$ and $u_1 = v_{x,1}/U_0$, with the subscripts $_{0,d,1}$ denoting values far upstream, at the disc and far downstream as indicated in Figure 2. Furthermore, $\boldsymbol{v} = \{v_s, v_n, v_\varphi\}$ is used, where $v_s = \sqrt{v_x^2 + v_r^2}$ is the velocity component along a streamline at the surface with constant $\Psi$, with $\Psi$ denoting the Stokes stream function.

The pressure and azimuthal velocity are discontinuous across the disc when $\epsilon \to 0$. For such an infinitely thin disc, integration of Eq. (1) yields:

$$\boldsymbol{F} = \lim_{\epsilon \to 0} \int_\epsilon \boldsymbol{f}\,dx = \boldsymbol{e}_x \Delta p + \boldsymbol{e}_\varphi \rho v_x \Delta v_\varphi, \tag{2}$$

where $\Delta$ denotes the jump across the disc, and $\boldsymbol{F}$ the applied surface load. A Joukowsky disc has a wake with swirl, induced by a vortex $\Gamma$ at the axis. The vortex core radius $\delta$ is assumed to be infinitely thin. The azimuthal velocity is:

$$v_\varphi = \frac{\Gamma}{2\pi r}. \tag{3}$$

The Bernoulli equation reads:

$$p + \frac{1}{2}\rho\boldsymbol{v}\cdot\boldsymbol{v} = H. \tag{4}$$

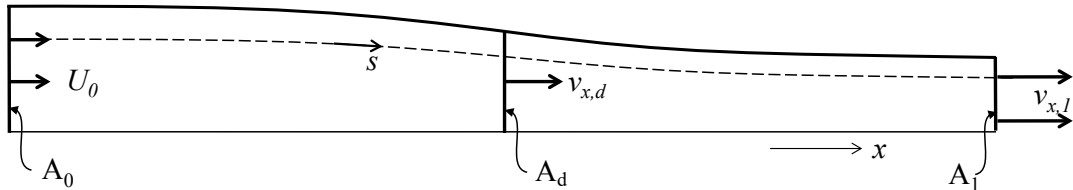

**Figure 2.** The stream tube of a propeller disc from cross sections $A_0$, infinitely far upstream, to $A_1$ in the fully developed wake. Only the upper half of the stream tube is shown.

When this is integrated across the disc and combined with Eq. (3), the axial component of Eq. (2) becomes:

$$F_x = \Delta p = \Delta H - \frac{1}{2}\rho \Delta v_\varphi^2 = \Delta H - \frac{1}{2}\rho \left(\frac{\Gamma}{2\pi r}\right)^2. \tag{5}$$

The power converted by an annulus $dr$ of the actuator disc equals the torque $Q$ times rotational speed $\Omega$ giving $\Omega dQ = 2\pi\Omega f_\varphi r^2 dr$, but also the integrated value of $\boldsymbol{f} \cdot \boldsymbol{v}$ with the use of Eq. (1), giving $2\pi r(\boldsymbol{v}.\boldsymbol{\nabla})H dr$. Equating both expressions shows that:

$$\boldsymbol{f} \cdot \boldsymbol{v} = \Omega r f_\varphi = (\boldsymbol{v}.\boldsymbol{\nabla})H. \tag{6}$$

$f_\varphi$ is expressed by the $\varphi$-component of the Euler equation (1): $f_\varphi = \rho(\boldsymbol{v} \cdot \boldsymbol{\nabla})v_\varphi$. Herewith:

$$\Omega r f_\varphi = \rho(\boldsymbol{v} \cdot \boldsymbol{\nabla})(\Omega r v_\varphi), \tag{7}$$

which gives with Eq. (6):

$$\frac{1}{\rho}\boldsymbol{\nabla}H = \boldsymbol{\nabla}(\Omega r v_\varphi) = \boldsymbol{\nabla}\left(\frac{\Omega\Gamma}{2\pi}\right). \tag{8}$$

Consequently, for a Joukowsky disc:

$$\Delta H = \rho\frac{\Omega\Gamma}{2\pi} = constant. \tag{9}$$

In the wind turbine mode $\Delta H < 0$, as energy is taken from the flow. With $\Omega$ always taken positive, $\Gamma$ and $v_\varphi$ are negative in the wind turbine mode, and positive in the propeller mode. This explains why $\Gamma_{axis}$ is shown with a negative sign in Fig. 1. Furthermore Eq. (9) shows that for $\Omega \to \infty$ meanwhile keeping $\Delta H$ constant, $\Gamma$ vanishes and, by Eq. (7), also $f_\varphi$. The result is the Froude disc without torque and wake swirl.

The power $P$ converted by the disc follows by integration of Eq. (6) on the actuator disc. In dimensionless notation this becomes:

$$C_p = \frac{1}{\frac{1}{2}\rho U_0^3 A_d}\int_A \boldsymbol{f} \cdot \boldsymbol{v} dA_d = \overline{u}_d\frac{\Delta H}{\frac{1}{2}\rho U_0^2} = 2\overline{u}_d\frac{\Omega R}{U_0}\frac{\Gamma}{2\pi RU_0}. \tag{10}$$

With $\lambda = \Omega R/U_0$ and $q = \Gamma/(2\pi RU_0)$, Eq. (9) becomes:

$$\frac{\Delta H}{\frac{1}{2}\rho U_0^2} = 2q\lambda, \tag{11}$$

and similarly Eq. (10):

$$C_p = 2q\lambda\overline{u}_d. \tag{12}$$

The thrust $T$ is derived in the same way, based on Eq. (5). Dimensionless, the thrust coefficient is $C_T = T/(\frac{1}{2}\rho U_0^2 A_d) = C_{T,\Delta H} + C_{T,\Delta v_\varphi}$ according to the two terms on the right-hand side of (5):

$$\left.\begin{array}{rcl} C_T & = & C_{T,\Delta H} + C_{T,\Delta v_\varphi} \\[2mm] C_{T,\Delta H} & = & 2\lambda q \\[2mm] C_{T,\Delta v_\varphi} & = & -q^2 \ln\left(\frac{R}{\delta}\right)^2 \end{array}\right\} \tag{13}$$

$C_{T,\Delta v_\varphi}$ does not contribute directly to the conversion of power, as it does not appear in Eq. (12). It is a conservative contribution to $C_T$, delivering the radial pressure gradient balancing the swirl immediately behind the disc. For finite $q$ and $\delta \to 0$,

$C_{T,\Delta v_\varphi} \to \infty$. For a non-zero $\delta$ combined with high $\lambda$, low $q$, $C_{T,\Delta v_\varphi}$ becomes small. For typical wind turbine parameters $\lambda = 8$, $C_{T,\Delta H} = -8/9$ and $\delta = 0.05R$ with $\delta$ representing the blade root cut-out area, $C_{T,\Delta v_\varphi} \approx -0.02$.

    The power and thrust have the same sign as $\Delta H$ or $q$: positive for propeller discs, negative for wind turbine discs. Consequently, the thrust and power (coefficients) are negative for discs extracting energy from the wake, and positive for discs adding energy to the wake.

The velocity in the far wake is characterised by $v_r = 0$. Herewith the Bernoulli equation (4) becomes in the far wake:

$$\frac{1}{\rho}(p_0 - p_1) = \frac{1}{2}\left(v_{x,1}^2 - U_0^2 + v_{\varphi,1}^2\right) - \Delta H. \tag{14}$$

The radial derivative is $\partial p_1/\partial r_1 = \rho(v_{\varphi,1}^2/r_1 - v_{x,1}\partial v_{x,1}/\partial r)$. When this is compared with the condition for radial pressure equilibrium in the fully developed wake, given by substitution of $v_r = 0$ in the radial component of Eq. (1):

$$\frac{\partial p_1}{\partial r_1} = \rho\frac{v_{\varphi,1}^2}{r_1}, \tag{15}$$

the result is $v_{x,1} = constant$ or, dimensionless, $u_1 = v_{x,1}/U_0 = constant$.

## 3   Flow pattern and average velocity

### 3.1   Momentum theory results for propeller and wind turbine discs

The momentum theory presented in van Kuik (2017) is valid when a different sign convention for $q$ is used, as in van Kuik (2017) it was defined $q = -\Gamma/(2\pi RU_0)$ instead of $\Gamma/(2\pi RU_0)$. This theory lacks an analytical solution. However, a numerical

solution of equation 29 of this paper is possible. Expressed in $\lambda$ and $q$, this is an implicit expression for $u_1$:

$$\frac{(1 - u_1)u_1^2 q^2}{1 - 2\lambda q - u_1^2} = \left(-q\lambda - \frac{1}{2}q^2\left(1 - \ln\left(\frac{q^2}{1 - 2\lambda q - u_1^2}\right)\right)\right), \tag{16}$$

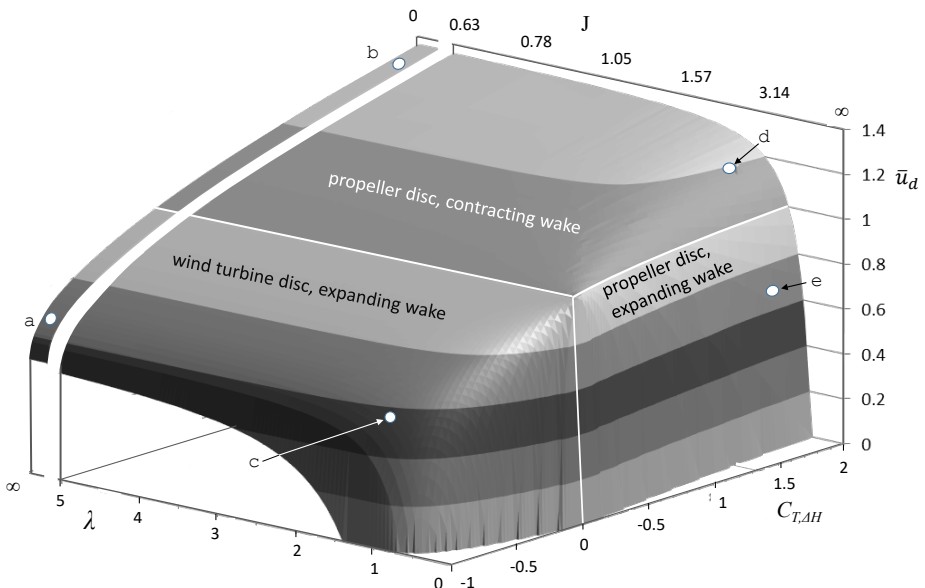

**Figure 3.** The axial velocity $\overline{u}_d$ for wind turbine discs ($-1 < C_{T,\Delta H} \leq 0$) and propeller discs ($0 \geq C_{T,\Delta H} < 2$) for $0 \leq \lambda \leq 5$ and for $\lambda = \infty$. The white markers a to e refer to flow cases defined in Table 1 and analysed in the next sections. The figure is a modified version of van Kuik (2018b, figure 6.2)

where $q$ has changed sign. After solving Eq. (16) for $u_1$, the wake expansion or contraction is given by van Kuik (2017, eq. 28). The average velocity at the disc $\overline{u}_d$ is given by van Kuik (2017, eq. 27), again with a change of sign of $q$ in both equations.

Figure 3 shows $\overline{u}_d$ for $-1 < C_{T,\Delta H} \leq +2$, $0 < \lambda \leq 5$ as well as $\lambda = \infty$. The advance ratio $J = \pi/\lambda$ is also given. The part of the figure with $C_{T,\Delta H} < 0$ shows $\overline{u}_d$ for wind turbine discs, with $C_{T,\Delta H} > 0$ for propeller discs. For $\lambda = 5$ the difference with $\lambda = \infty$ is smaller than $0.7\,\%$ so the Froude momentum theory results are practically recovered. Apparently, swirl has little effect when $\lambda > 5$. The flow cases a to e are defined in Table 1, together with two flow parameters: the dimensionless average velocity at the disc, $\overline{u}_d$ and the dimensionless absolute velocity in the meridian plane $|\boldsymbol{v}|_m = \sqrt{v_{x,d}^2 + v_{r,d}^2}/U_0$. $|\boldsymbol{v}|_m$ is the same as the velocity along a streamline $v_s$ at the position of the disc, so $|\boldsymbol{v}|_m = v_{s,d}/U_0$. $\overline{u}_d$ and $|\boldsymbol{v}|_m$ and will be examined in

the next sections.

**Table 1.** Definition of actuator disc flow cases a to e, the average velocity at the disc $\overline{u}_d$ and the absolute velocity in the meridian plane $|\boldsymbol{v}|_m$.

|  |  | $C_{T,\Delta H} = -8/9$ | | $C_{T,\Delta H} = 16/9$ | |  |
|---|---|---|---|---|---|---|
|  |  | $\overline{u}_d$ | $|\boldsymbol{v}|_m$ | $\overline{u}_d$ | $|\boldsymbol{v}|_m$ |  |
| $\lambda = \infty$ | a: | 0.666 | 0.684 | b: 1.333 | 1.348 | $J = 0$ |
| 1 | c: | 0.553 | 0.588 | d: 1.195 | 1.197 | $\pi$ |
| 0.5 |  |  |  | e: 0.679 | 0.712 | $2\pi$ |

Several particularities can be observed in Fig. 3:

- For values of $\lambda < 1.4$ the minimum attainable $C_{\Delta H} > -1$, giving $\overline{u}_d = 0$, so the flow is blocked. Such a minimum $\lambda$ exists in the wind turbine as well as propeller flow regime.

- For wind turbine discs having $\lambda > 1.4$ the minimum $C_{\Delta H}$ is $-1.0$, with $\overline{u}_d$ shrinking from 0.5 at $\lambda = 5$ to 0 at $\lambda \approx 1.4$.

- For propeller discs having a very high $J$, $\overline{u}_d < 1$ so the wake expands. This upper boundary of the expanding wake region is the line $\overline{u}_d = 1$, giving an undeformed wake. The lower boundary is defined by $\overline{u}_d = 0$, giving blocked flow. Both boundaries put a limit to the maximum attainable $C_{T,\Delta H}$. For low $J$ there is no upper limit for $C_{T,\Delta H}$: the wake can be accelerated to any value.

These particularities will be discussed in the next subsections, to start with the propeller disc.

## 3.2 Propeller discs having an expanding wake

For low rotational speed (low $\lambda$, high $J$), the average axial velocity at the disc $\overline{u}_d$ deviates from the famous Froude result: $\overline{u}_d < \frac{1}{2}(u_1 + 1)$. This happens in both flow regimes. Responsible for this is the radial pressure distribution necessary to maintain the swirl. This gives a contribution to the momentum balance, as is explained in van Kuik (2018b, chapter 6). The first term in the disc load Eq. (5) gives the contribution of $\Delta H$ to the disc load, the second term the swirl related pressure contribution. This contribution $-\frac{\rho}{2}(\Gamma/(2\pi r))^2$ is always $< 0$, while the sign of the first term depends on the actuator disc mode: for wind turbine discs $< 0$, for propeller discs $> 0$. Consequently, both terms may cancel for propeller flows, resulting in a zero pressure jump at $r = R$. With Eqs. (5) and (9) the condition for this particular flow is derived: $\Omega R = -\frac{1}{2}v_\varphi$ or:

$$\lambda = q/2. \tag{17}$$

In Fig. 3 this specific flow regime is indicated by the line separating the propeller disc regime with a contracting wake, from the propeller disc regime with an expanding wake, with $\overline{u}_d = 1$ at the separation line. The resulting flow has a wake with constant radius, so $v_x = U_0$, $v_r = 0$ throughout the flow. In the wake $v_\varphi = \Gamma/(2\pi r)$. The vortex sheet separating the wake from the outer flow consists of axial vorticity across which $\Delta H = \frac{1}{2}(\Omega R)^2$.

For lower rotational speeds the pressure jump at the edge has become $< 0$, as the swirl-related pressure term in Eq. (5) overrules the $\Delta H$ term, thereby generating wake boundary vorticity as for wind turbine disc flows. Although kinetic energy in the wake is lower than outside the wake, the disc load adds potential energy (pressure) to the flow such that the total energy in the wake is higher than upstream. More explanation of this remarkable flow regime is provided in van Kuik (2018b, section 6.3).

## 3.3 Minimum $\lambda$ operation with blocked flow

In van Kuik (2018b, section 6.3) the operation at minimum possible $\lambda$ is analysed. In this flow case $\overline{u}_d = 0$ as well as $u_1 = 0$, so the disc acts as a blockage to the flow. In the wake the change of axial momentum is zero, but $H_{wake} - H_0 \neq 0$ as the azimuthal velocity is non-zero. Lower values of $\lambda$ are not possible.

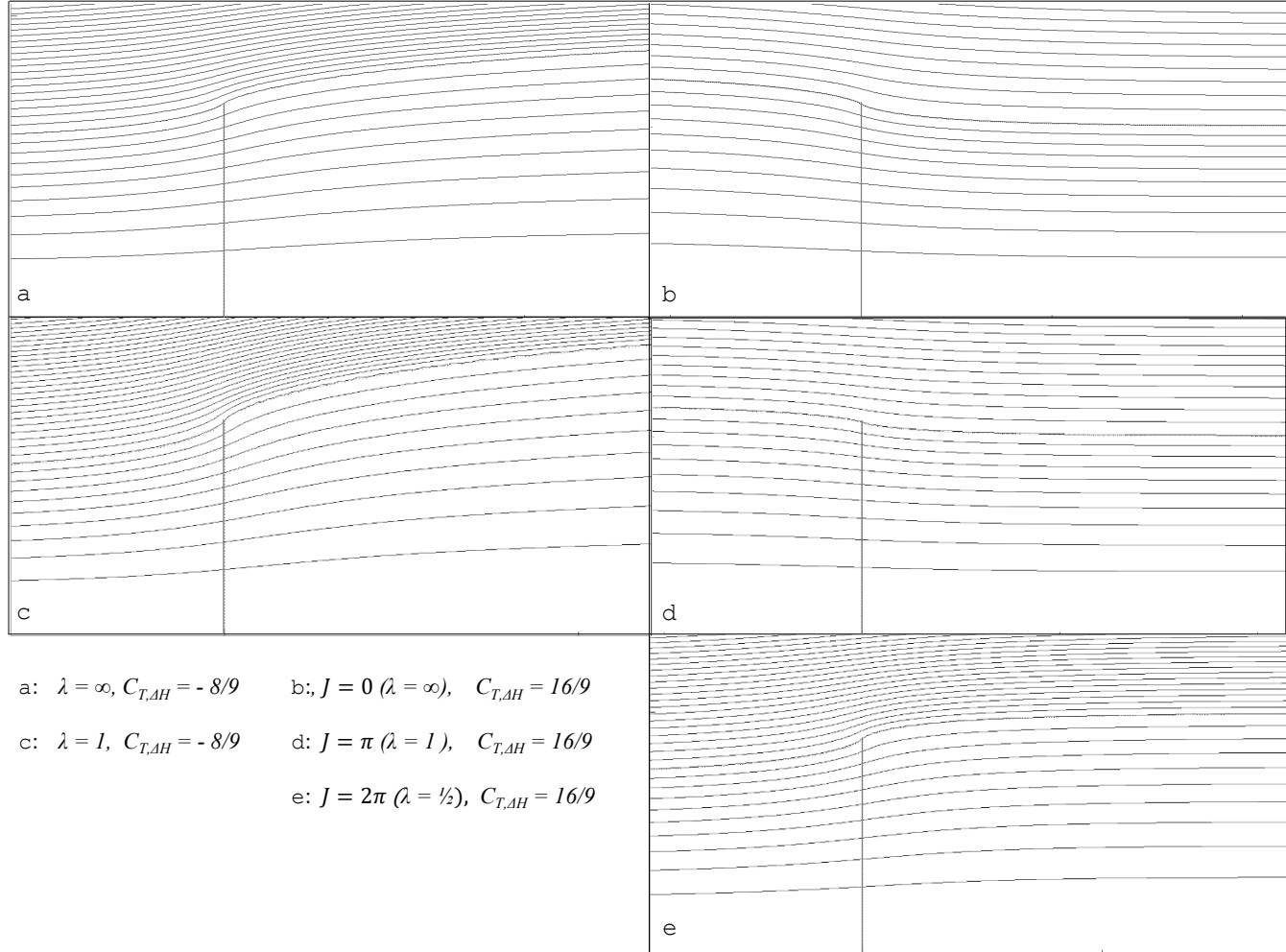

a: $\lambda = \infty$, $C_{T,\varDelta H} = -8/9$     b:, $J = 0$ ($\lambda = \infty$),   $C_{T,\varDelta H} = 16/9$

c: $\lambda = 1$, $C_{T,\varDelta H} = -8/9$     d: $J = \pi$ ($\lambda = 1$),   $C_{T,\varDelta H} = 16/9$

e: $J = 2\pi$ ($\lambda = \frac{1}{2}$), $C_{T,\varDelta H} = 16/9$

**Figure 4.** The flow patterns of wind turbine discs a and c, and propeller discs b and d with a contracting wake, e with an expanding wake. The streamlines indicate stream tube values increasing with $\Delta\Psi = 0.1\Psi_1$.

## 3.4 Flow patterns

Table 1 show the flow cases, also indiated in Fig. 3, for which the flow field has been calculated numerically with the potential flow method used in van Kuik (2017). An assessment of the accuracy presented in van Kuik (2018b, appendix D). The highest attainable accuracy is applied: calculated values of integrated properties like wake expansion of contraction deviate less than 0.3 % from momentum theory values. The same holds for the local satisfaction of the boundary conditions at the wake boundary $v_n = 0, \Delta p = 0$, except within a distance $0.02R$ from the disc edge, where $v_n$ may deviate up to $0.02U_0$ without challenging the condition $\Psi = \Psi_1$ and without affecting integrated flow quantities.

In Fig. 4 the streamlines of flow cases a to e are shown, grouped according to their position in Fig. 3. Flow cases a looks similar to flow case e, although the latter is a propeller disc flow.

## 4 The velocity distribution at the disc

With $v_s$ being the velocity in the meridian plane, $v_s/U_0$ at the upstream side of the disc equals $|v|_m = \sqrt{v_{x,d}^2 + v_{r,d}^2}/U_0$. Table 1 gives the numerical values of $\overline{u}_d$ and $|v|_m$ for the flow cases considered. The differences between $\overline{u}_d$ as calculated numerically and as resulting from the momentum theory is $0.2\%$ or less. The value for $|v|_m$ is the value for $r = 0$. Figure 5 shows the distribution of the axial and radial velocity components and the meridional velocity. Most striking is the distribution of this meridional velocity being practically uniform. The explanation of this is presented in section 5, but first the velocity distributions shown in Fig. 5 are analysed.

### 4.1 The meridional velocity

Fig. 5 shows the amount of non-uniformity in $|v|_m$. This non-uniformity is defined as $|v|_m(0.97)/|v|_m(0) - 1$, expressed in percentages, except for flow case a. In all flow cases except a, $|v|_m$ increases or decreases monotonically from $r = 0$ towards $r = R$. In flow case a, $|v|_m$ increases with increasing $r$, with the maximum, $0.2$ %, reached at $r/R = 0.8$ after which it decreases towards the disc edge. At $r/R = 0.97$, $|v|_m$ differs $-0.1$ % from its value at $r/R = 0$, so for a the non-uniformity number indicates $|v|_m(0.8)/|v|_m(0) - 1$. These numbers for a are within the uncertainty range of the calculations, so their significance is not clear. The choice for $r = 0.97R$ in the other flow cases is somewhat arbitrary, but is motivated by the argument that the sharp transition at $r/R = 1$ shown in Fig. 5, is not physically realistic. Viscosity will smooth this transition depending on the Reynolds number used, as shown in Sørensen et al. (1998).

#### 4.1.1 Wind turbine flows

As shown in Fig. 5 $|v|_m$ is practically uniform in flow case a: the non-uniformity is $-0.2$ %. For low $\lambda$ operation the non-uniformity is stronger: $-1.8$ % for flow case c. The non-uniformity is checked (but not shown in a figure) for several other flow cases:

- Disc load $C_{\Delta H} = -8/9, \lambda = 5$ instead of $\infty$: the result differ less than $0.1$ %.

- Discs with $\lambda = \infty$ but heavier disc loads: the non-uniformity in $|v|_{d-}$ is $-0.7$ % for $C_{T,\Delta H} = -0.97$, $-0.8$ % for $C_{T,\Delta H} = -0.995$.

The optimal operational regime of modern wind turbines is $\lambda > 5$ with $C_{T_{\Delta H}} > -0.9$, so the non-uniformity in $|v|_m$ of flow cases representing this optimal regime is negligible.

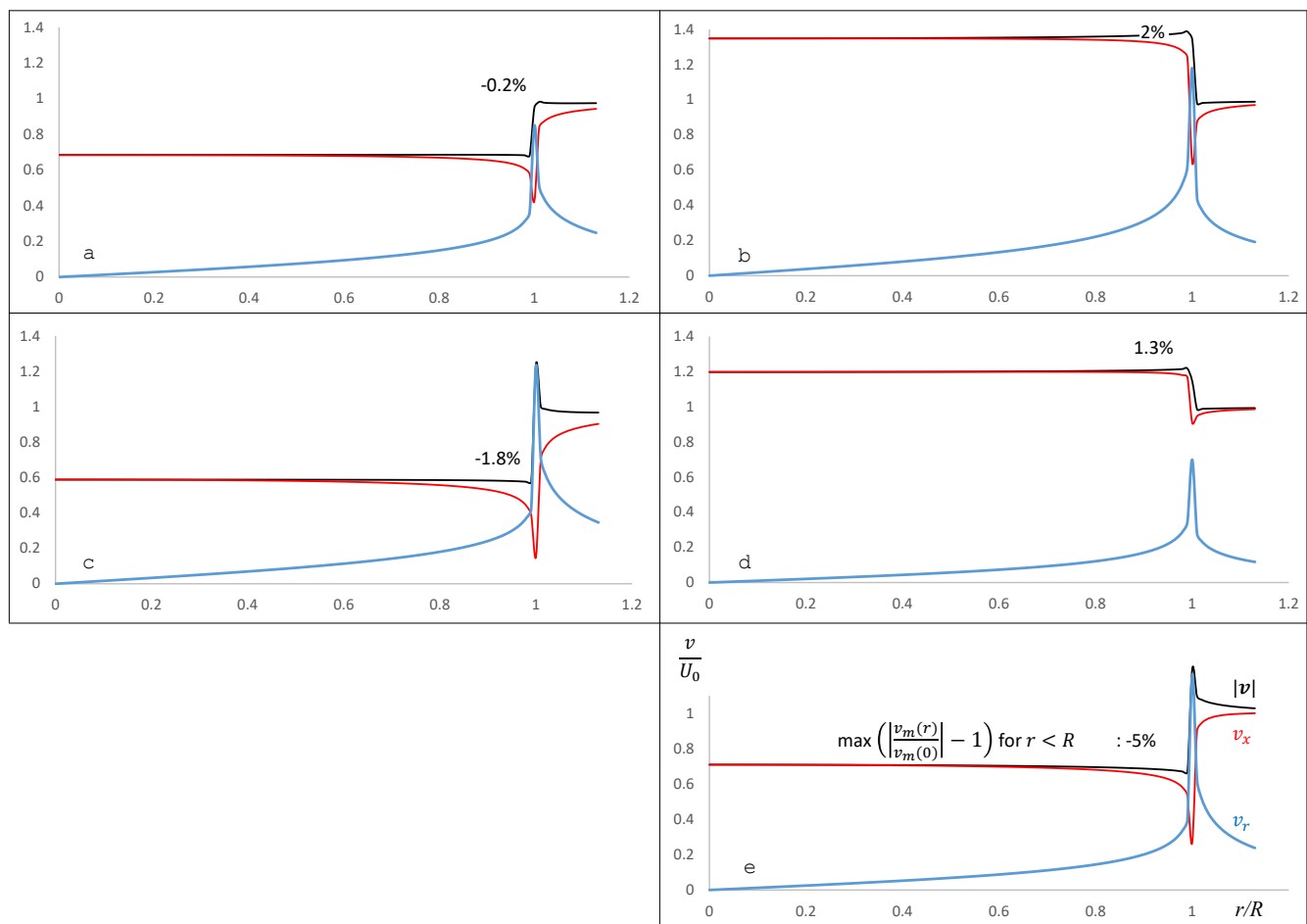

**Figure 5.** The velocity distribution at the disc, for flow cases a to e defined in Fig. 4. Black line: $|\boldsymbol{v}|_m = \sqrt{v_x^2 + v_r^2}/U_0$, red line: $u = v_x/U_0$, blue line $v_r/U_0$. All vertical axes have the same scale. The percentages denoting the non-uniformity of $|\boldsymbol{v}|_m$ are explained in section 4.1.

### 4.1.2 Propeller flows

The non-uniformity in $|\boldsymbol{v}|_m$ is 2 % in flow case b, $J = 0$. It decreases to 1.3 % in flow case d, $J = \pi$, becomes 0 for $J = 1.5\pi$ when the flow case without wake deformation is reached according to Eq. (17), and becomes strongly negative for higher $J$ as shown in flow case e: $-5$ % for $J = 2\pi$. Usually the advance ratio $J$ is lower than 2.5, see for example McCormick (1994, figure 6.12). Fig. 3 shows that in this regime the impact of wake swirl is very limited, so flow case b is considered representative, with a non-uniformity of $\approx 2$ %.

## 4.2 The axial velocity

In all flow cases the axial velocity is far from uniform, as was already shown by e.g. Sørensen et al. (1998), Madsen et al. (2010). For Froude wind turbine discs, the cause of this has been addressed in van Kuik and Lignarolo (2016), for Joukowsky disc flows in van Kuik (2017). In terms of the momentum balance, the source of this non-uniformity is the pressure acting on the sides of a stream annulus used as control volume. When the stream tube boundary is used as boundary of the control volume, the pressure at this boundary does not give a contribution in axial direction, but for stream annuli this is not the case. When this pressure is calculated and included in the momentum balance, the prediction of $u_d$ per annulus by the momentum theory matches the calculated, non-uniform distribution of the $u_d$. This may serve as the explanation of the non-uniformity of $u_d$, but cannot be used as a prediction model, as the pressure is not known a priori. For Froude discs the $u_d$ distribution has been calculated for $-1 < C_{T,\Delta H} < 0$ enabling a surface-fit engineering approximation for $u_d(\frac{r}{R}, C_{T,\Delta H})$, see van Kuik and Lignarolo (2016, section 5.2).

## 4.3 The radial velocity

The radial velocity receives little attention in actuator disc and rotor publications compared to the axial velocity. Some exceptions are Madsen et al. (2010), presenting an engineering model for the decreased axial velocity close to the disc or rotor edge based on the radial velocity, Micallef et al. (2013), comparing calculated and measured radial velocity near rotor blade tips to asses blade bound chordwise vorticity in order to explain the initially inward motion of the tip vortex, and van Kuik et al. (2014, section 4), quantifying this chordwise vorticity and the associated tip load responsible for this inward tip vortex motion, and Sørensen (2015, section 3.2), analysing $\partial v_r/\partial x$ at the plane of the disc.

Recently Limacher and Wood (2019) found a relation between the axial and radial velocity component in the rotor or disc plane:

$$\int_{S_d} \left(\frac{v_{r,d}}{U_0}\right)^2 - \left(1 - \frac{v_{x,d}}{U_0}\right)^2 dS = 0. \tag{18}$$

where $S_d$ is the plane of the disc or rotor from $r = 0$ to $r = \infty$. The quantity $1 - v_{x,d}/U_0$ is known as the induction $a$. Based on (18) Limacher and Wood (2019) conclude that $v_{r,d}/U_0$ and $a$ have to be equal close to the disc edge or rotor tip, so:

$$\left.\begin{aligned} \frac{v_{x,d}}{U_0} + \frac{v_{r,d}}{U_0} &= 1 \qquad \text{for flows with wake expansion} \\ \frac{v_{x,d}}{U_0} - \frac{v_{r,d}}{U_0} &= 1 \qquad \text{for flows with wake contraction} \end{aligned}\right\} \text{ at } r \approx R \tag{19}$$

Eqs. (18) and (19) have been evaluated using the velocity distributions of Fig. 5. For flow case a the left-hand side of Eq. (18) indeed approaches 0 for increasing radius of $S_d$. Table 2 gives the radial coordinate where (19) is satisfied: almost at the disc edge for the flow states with an expanding wake a, c, e, while flow states b, d with a contracting wake show this property at a smaller radius. The expanding flows exhibit steep changes in $v_x$ and $v_r$ close to $r = R$. An exact assessment the radial position where Eq. (19) is satisfied is difficult for which reason a range of $r$ is given.

**Table 2.** The radial position $r/R$ where (19) is satisfied.

| | $(v_r + v_x)_d = U_0$ | | $(v_r - v_x)_d = U_0$ |
|---|---|---|---|
| a: | $0.99 < r/R < 1$ | b: | $0.912$ |
| c: | $0.99 < r/R < 1$ | d: | $0.932$ |
| e: | $0.99 < r/R < 1$ | | |

Eq. (19) provides a second relation between $v_{x,d}$ and $v_{r,d}$, besides the conclusion of section 4.1 that $|\boldsymbol{v}|_m$ is practically constant for $r < R$. This allows an engineering estimate of the wake expansion at the disc for wind turbine flows, when it is assumed that $|\boldsymbol{v}|_m = \sqrt{v_{x,d}^2 + v_{r,d}^2}/U_0 = $ constant and $v_{x,d} + v_{r,d} = U_0$ at $r/R = 1$. As an example the flow with $v_{x,d} = v_{r,d}$ is evaluated, giving $|\boldsymbol{v}|_m = 0.707$ and $v_{x,d} = v_{r,d} = 0.5U_0$ at $r = R$. This gives a slope of the vortex sheet shape of $45^0$ at $r = R$.

235  This is close to flow state a, where the numerically calculated slope is $46^0$, and $|\boldsymbol{v}|_m = 0.684$ which is $3,3\%$ lower than the estimate. Further exploration of such an engineering estimate is left for future work.

## 5  Explanation of the (non-)uniformity of $|v|_m$

The Euler equation of motion (1) offers a first-order explanation for the observation that $|\boldsymbol{v}|_m$ is practically uniform for $\lambda \geq 1$. The radial component of Eq. (1) reads:

240
$$\frac{\partial p}{\partial r} = -\rho v_s \frac{\partial v_r}{\partial s} + \frac{v_\varphi^2}{r}. \tag{20}$$

Equation (3) for $v_\varphi$ combined with Bernoulli's equation (4) gives a second equation for $\partial p/\partial r$:

$$\frac{\partial p}{\partial r} = -\rho v_s \frac{\partial v_s}{\partial r} + \frac{v_\varphi^2}{r}, \tag{21}$$

so the result is $\partial v_s/\partial r = \partial v_r/\partial s$, or at the disc :

$$\frac{\partial v_{s,d}}{\partial r} = \frac{\partial v_{r,d}}{\partial s}. \tag{22}$$

245  Consequently, the distribution of $v_{s,d}$ is determined by the derivative $\partial v_r/\partial s$ along the streamline. In case $v_r$ has a maximum or minimum at the disc, $v_{s,d}/U_0 = |\boldsymbol{v}|_m$ is uniform.

Qualitative observations regarding the in- or decrease of $v_r$ are possible when moving the position along a streamline in the meridian plane. The radial velocity depends only on the vorticity $\gamma_\varphi$ distributed along the wake boundary, and the position of obervation $s^*$. For a disc with an expanding wake, the following relations hold:

250    a) At the upwind side of the streamline: when moving towards the disc, the distance to $\gamma_\varphi$ decreases, so $v_r$ increases, and $\partial v_r/\partial s > 0$.

b) At the downwind side of the disc the streamline is to be distinguished in two parts: upstream and downstream of $s^*$. The upstream vorticity induces a negative $v_{r,upstream}$, becoming more negative when $s^*$ moves downstream, leading to

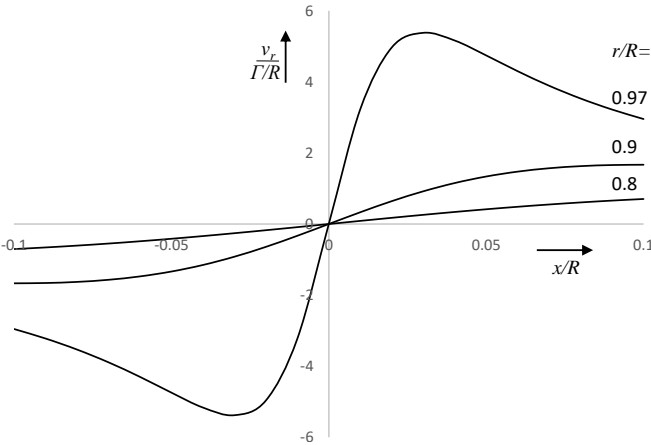

**Figure 6.** The radial velocity induced by a unit vortex ring positioned at $x = 0, R = 1$, at the lines $r/R = 0.8, 0.9, 0.97$ .

$\partial v_{r,upstream}/\partial s < 0$. The part of the wake downstream of $s^*$ remains a semi-infinite wake, so $v_{r,downstream}$ is expected
to vary only little for increasing $s^*$ (this is to be verified later), leading to $\partial v_{r,downstream}/\partial s \approx 0$. This gives for the total
induction in the wake $\partial v_r/\partial s < 0$.

Consequently, according to a) and b) $\partial v_r/\partial s = 0$ at the disc and with Eq. (22) $\partial v_{s,d}/\partial r = 0$ so $|\boldsymbol{v}|_m$ is uniform.

For flow cases with a contracting wake the same reasoning is valid, with an appropriate change of signs, leading to a
minimum $v_r$ at the disc and a uniform $|\boldsymbol{v}|_m$.

However, these qualitative considerations miss the effect that a vortex ring induces a non-zero $\partial v_r/\partial s$ in the plane of
the ring. Figure 6 shows the calculated radial velocity induced by a vortex ring positioned at $x = 0, R = 1$ along the lines
$r/R = 0.8, 0.9, 0.97$. The shape of the plot resembles the induction $v_r = \frac{\Gamma}{2\pi} \frac{cos\alpha}{dist}$ by a point vortex in a $2 - D$ plane, where $dist$
is the distance to the vortex, and $\alpha$ the angle the angular coordinate around the vortex position. As is clear by Fig. 6, this effect
is strongest close to the position of the ring, as $\partial v_r/\partial x \to \infty$ for $r/R \to 1$. Apart from the distance to the ring, the strength of
the ring determines the local value of $\partial v_r/\partial x$, as its value is linear in this strength.

For a vorticity tube things are slightly different, as is easily shown by the example of a tube of constant strength with a semi-
finite length. Each elementary vortex ring $\gamma dx$ induces a non-zero $\partial v_r/\partial r$ in its own plane, but due to symmetry considerations
this is annihilated except near and at the beginning of the tube. Also for the vorticity tube surrounding the actuator disc wake,
the singular behaviour of $\partial v_r/\partial s$ is annihilated everywhere by the induction of upstream and downstream vorticity, except at
the leading edge of the wake. There the sign of the contribution to $\partial v_r/\partial r$ at $x = 0$ is opposite to the sign of $\partial v_r/\partial r$ at $x < 0$,
as is clear from Fig. 6.

The considerations a) plus b) mentioned above include the effect of increasing or decreasing distance to the vorticity and the
change of sign of $v_r$ for vorticity upstream and downstream of $s^*$, but the non-zero $\partial v_r/\partial s$ at $s^* = s_d$ has to be added:

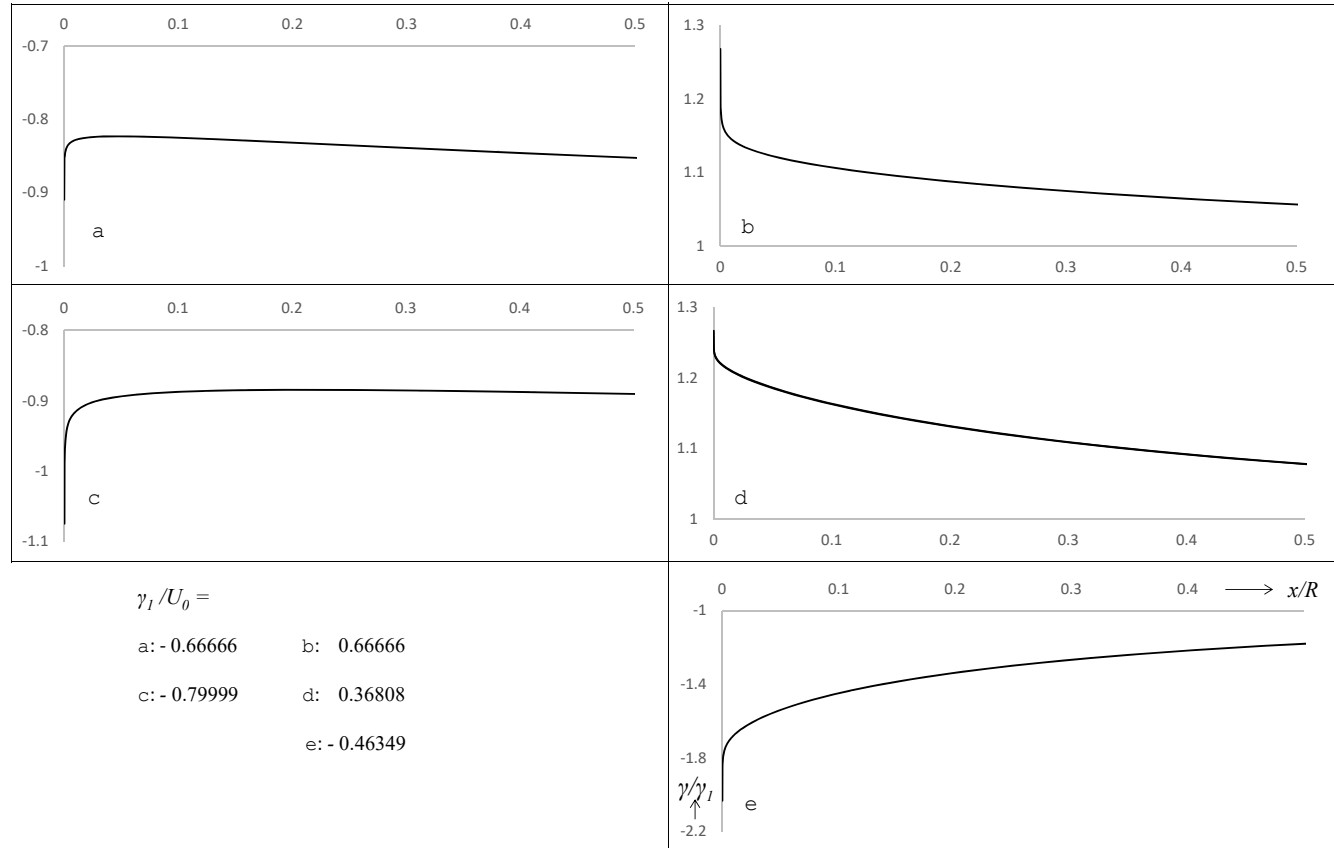

**Figure 7.** The distribution of the vortex sheet strength $\gamma_\varphi(x)/\gamma_{\varphi,1}$, for flow cases a to e, defined in Fig. 4. The vertical axes have the same scale, except the axis of e, which covers a 4 times larger range of $\gamma$.

c) At $s^* = s_d$ the induction by the leading edge vorticity at the disc edge adds a contribution to $\partial v_r/\partial s$ depending on the local vorticity strength and the inverse of the distance to the disc edge. The sign of the contribution is opposite to the sign of $\partial v_r/\partial s$ upstream of $s_d$.

d) According to a) and b), the position where $\partial v_r/\partial s = 0$, is at the disc. With c) it moves upstream of the disc, for all disc flows. How far it moves upstream depends on the strength of the leading edge vorticity. For discs with an expanding wake, using Eq. (22), $\partial v_{r,d}/\partial s = \partial v_{s,d}/\partial r < 0$, for discs with a contracting wake $\partial v_{r,d}/\partial s = \partial v_{s,d}/\partial r > 0$. This is in agreement with Fig. 5, showing that $v_{s,d}$ diminishes towards $r = R$ for flow cases a, c, e, while it increases for flow cases b, d.

This qualitative line of arguments a) - d) requires a numerical validation and quantification. The calculated wake vorticity $\gamma_\varphi(x)/\gamma_{\varphi,1}$ is shown in Fig. 7, with $\gamma_{\varphi,1}$ being the azimuthal vorticity in the far wake: $\gamma_{\varphi,1} = v_{x,1} - U_0$. In all flow cases the distributions have a singularity at the leading edge. Flow case a has the weakest singularity, flow case e the strongest. Fig.

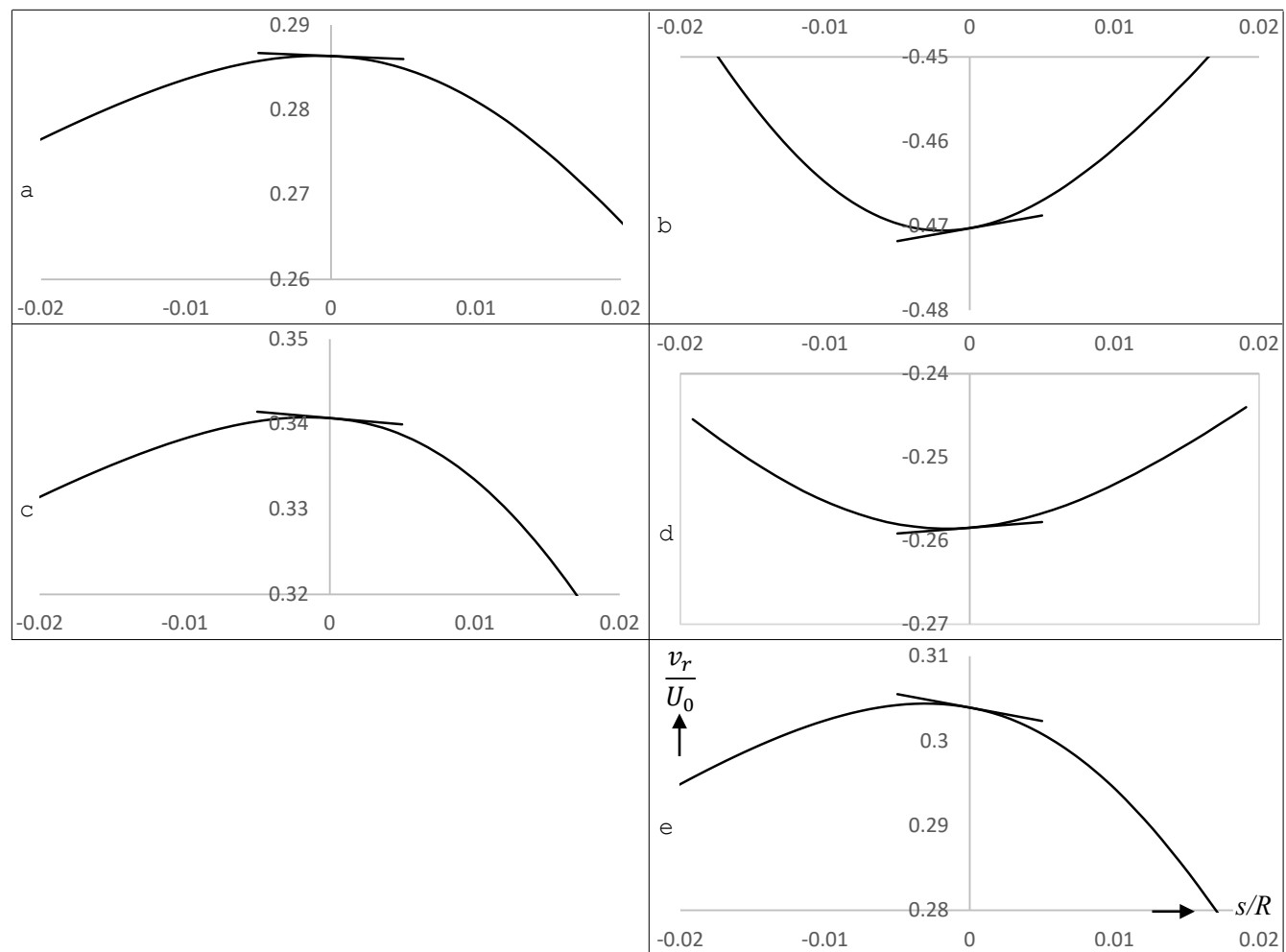

**Figure 8.** Curved lines: the radial velocity along the streamline passing the disc at $r/R = 0.97$, straight lines: the tangent of the distribution $v_{s,d}(r)/U_0$ at $r/R = 0.97$, plotted through the $s = 0$ position at the curved line.

285 8 shows the calculated $v_r$ along a streamline passing the disc at $r/R = 0.97$ (curved lines) and the tangent at $r/R = 0.97$ of the distribution $v_m(r)$ (straight line), plotted through the $s = 0$ position at the curved line. As is clear from the graphs, these straight lines coincide with the tangents to the $v_r(s)$ distribution, confirming Eq. (22). Furthermore, downstream of the disc $v_r$ decreases for flow cases a, c, e and increases for b, d, thereby confirming the assumption made in b).

The absolute value of the slope of the tangents is lowest in flow case a, highest in e. This is in agreement with the strength
290 of the leading edge singularity of $\gamma_\varphi(x)/\gamma_{\varphi,1}$ and the non-uniformity of $v_m$. In all flow cases $v_r(s)$ reaches a maximum or minimum just upstream of the disc: at $s/R = -0.00155$ for a, and $-0.00252$ for e, with the values for other flow cases in between these positions.

## 6 Conclusions

With respect to the average velocity at the actuator disc:

- For Joukowsky discs in wind turbines and propellers mode, the average velocity has been found, from $\lambda = 0$ up to $\lambda \to \infty$, or $J \to \infty$ to $J = 0$.

- For a very high $J$, propeller disc flows have an expanding wake while still energy is put into the wake. The high angular momentum of the wake flow creates a pressure deficit in the wake, which is supplemented by the pressure added by the disc. This results in a positive energy balance while the wake axial velocity has gone down.

- Propeller discs flows without wake expansion or contraction are possible for specific values of $J$, marking the transition from the contracting wake operational mode at low $J$, to the expanding wake mode at high $J$.

- In the propeller as well as wind tubine flow regimes the velocity at the disc becomes $0$ for very low rotational speed, resulting in a flow with a blocked disc.

With respect to the distribution of the velocity in the meridian plane at the disc position:

- $|\boldsymbol{v}|_m$ is practically uniform for wind turbine disc flows with $\lambda > 5$ (deviation in the order of a few ‰)

- $|\boldsymbol{v}|_m$ is almost uniform for wind turbine disc flows with low $\lambda$ and propeller flows with $J \approx \pi$ (deviation in the order of a few %)

- $|\boldsymbol{v}|_m$ is non-uniform for the propeller disc flow with wake expansion at very high $J$ (deviation in the order of several %).

- the differences in uniformity are caused by the different strengths of the leading edge singularity in the wake boundary vorticity strength.

*Acknowledgements.* The author thanks the two reviewers, David Wood and an anonymous expert, as their comments improved the manuscript significantly. The same holds for the discussion with David Wood, University of Calgary, Canada, and Eric Limacher, Federal University of Pará, Belém, Brazil, about the significance of equations (18) and (19).

**Data availability** The dataset "Background data for On the velocity at wind turbine and propeller actuator discs, WES-2020-51", van Kuik (2020), will be stored at the repository of the Dutch Universities of Technology, http://researchdata.4tu.nl/home/. The DOI will be given in the final version.

**Competing interests** The author declares that he has no conflict of interest.

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
