# Peer review of "On the velocity at wind turbine and propeller actuator discs"

_Wind Energy Science, 2020_

## Referee Comment (RC1) · David Wood (Referee) · 22 Mar 2020

Fundamental studies of wind turbines have fundamental importance and so I welcome this new instalment of the author's long involvement with basic aerodynamic issues for horizontal-axis wind turbines and propellers in terms of actuator disks. The classification of flow regimes shown in figure 4 – which is taken from van Kuik (2018b, reference in the manuscript) - provides a link between propellers and turbines and has some counter-intuitive features, such as wake expansion for propellers adding energy to the flow, and a minimum power producing tip speed ratio for wind turbines. What is new in this contribution is the analysis of the velocities through the disk which provides new and important information. I am particularly interested in the radial velocity, which has received little attention in wind energy research, apart from Madsen et al. (2010, reference below), Micallef et al. (2013), and Sørensen (2015, reference in manuscript). Limacher & Wood (2020, reference below) showed that $v^2 - a^2$, where $v$ is the normalized radial velocity and $a$ is the axial induction factor, integrates to zero over the radial plane including the disk and argued that this requires $v \approx a$ at the edge of the disk, which appears to be consistent with the results in figure 9.

I did not find any errors in the analysis, the writing, though terse, is good, and the figures are of high quality. My only recommendations are: (a) that the link between the author's previous work be made clearer – as implied by my comment on figure 4 – and (b) that the previous work on the radial velocity be discussed. My hope is that this work will inspire measurements of these flow regimes that are not of direct interest for modern, power producing turbines, but are fundamental to our understanding of how wind turbines work.

Additional References

Limacher, E. J., & Wood, D. H. (2020) Derivation of an Impulse Equation for Wind Turbine Thrust. Wind Energy Science (submitted) https://www.wind-energ-sci-discuss.net/wes-2019-93/

Madsen, H. A., Bak, C., Døssing, M., Mikkelsen, R., Øye, S.: Validation and modification of the blade element momentum theory based on comparisons with actuator disc simulations. Wind Energy, 13(4), 373-389, 2010.

Micallef, D., van Bussel, G., Ferreira, C. S., & Sant, T. (2013). An investigation of radial velocities for a horizontal axis wind turbine in axial and yawed flows. Wind Energy, 16(4), 529-544.

---

## Referee Comment (RC2) · Anonymous Referee #2 · 28 Apr 2020

The article concerns the analysis of the flow on an actuator disc, with the aim of understanding and analyzing the velocity distribution on the disc when applying the axisymmetric Joukowsky rotor model for both wind turbine and propeller states. The background for the Joukowsky model is that it assumes a constant circulation on the rotor disc, which makes the model very convenient as a simple means for rotor analysis. On the other hand, it may introduce some unwanted properties at small tip speed ratios (or high advancing ratios) if not treated properly. This has been the issue for discussions and the author has in earlier papers contributed to the understanding of the model. The present work is based on and is a continuation of the previous work, with the main original contribution being the understanding of the velocity distribution at small tip speed ratios and the difference observed when comparing wind turbine flows with propellers. The introduction gives an overview of previous work in order to explain the news of the present article, which among others thing include new computations of higher accuracy than what has previously been presented.

The overall analysis and conclusion are easy to understand. In particular, the analytical analysis in section 5 and its support by numerical results is excellent. Based on this, I have no problems in accepting the paper. However, there are some issues, shown below, I will ask authors to clarify:

• I don't understand the sentence (p6, line 146-149): 'The resulting flow has wake with constant radius, so vx = Uo, vr = 0 throughout the flow. In the wake v $\phi = \Gamma/(2\pi r)$ . The vortex sheet separating the wake from the outer flow consists of axial vorticity across which  $\Delta H = \frac{1}{2} (\Omega R)^2$ . The line ud = 1 in Fig. 3 indicates this flow state.'

What is written is not a flow state, but just the difference in vorticity between outside and inside of the wake. How can this be a flow state? And how can this be related to Fig.3? Please explain.

- I find it difficult to relate Fig. 4 to Fig.3. If it is a horizontal cut, then Ud must be a constant, which I doubt. If it is not horizontal, I suggest that the Ud-value is given for the various cases (a-e) to help the reader to better understand the figure. I fully aware that this not important as long as the different a-e situations represent different significant cases. But it can be somewhat confusing for understanding the figure.
- Page 8 line 159: Here you introduce a  $\gamma 1/U0 = -2/3$ , but we don't know what  $\gamma 1$  is and why this is important. Would it not be more correct to show Ud/Uo. Or is this the same? Please explain.

---

## Editor Comment (EC1) · Alessandro Bianchini (Editor) · 29 Apr 2020

Dear Prof. van Kuik, your paper received positive comments. I encourage you to reply to the Reviewers' comments and to incorporate their suggestions into a new version of your paper. Best regards,

Alessandro Bianchini

---

## Editor Comment (EC2) · Alessandro Bianchini (Editor) · 19 May 2020

Dear Prof. van Kuik, I have gone through your responses to the Reviewers' comments. On this basis, I encourage you to submit officially the revised version of your study. Best regards

---

## Author Comment (AC1) · 19 May 2020

The author thanks both reviewers for their positive as well as critical remarks. These have led to a real improvement of the paper, and to an extension with a paragraph on the radial velocity component. Here you find my reply to the comments of both reviewers. I have uploaded a new version of the manuscript including indications of the changes in text and figures (please activate the 'comments' tab).

Reviewer RC1: 1: The reviewer asks attention for the radial velocity, with suggestions for additional literature. Reply: A subsection '4.3 The radial velocity' has been added, and the radial velocity component is now shown too in Fig. 5. The reviewer mentions a specific relation between the axial and radial velocity, derived in a manuscript-fordiscussion by him and a co-author. This relation is mentioned, analysed and confirmed in section 4.3. 2: Furthermore the reviewer likes to see clearer the link with the author's previous work, specifically w.r.t. (previous) Figure 4. Reply : This figure has been deleted. The information considered essential for the line of thoughts is now included in a new version of Figure 3. This figure shows the flow regimes and the flow cases a-e. Moreover, the caption of this figure mentions the original figure in a previous publication.

Reviewer RC2: 1: The reviewer likes a clarification of the paragraph right after eq. (17), about the propeller flow without wake expansion of contraction. Especially the word 'flow state' was confusing. Reply: the words 'flow state' are not used any more. Instead the actuator disc flows that have been analysed in detail are called 'flow cases'. The paragraph right after eq. (17) has been adapted accordingly. 2: The reviewer finds it difficult to relate (former) Figure 4 to Figure 3. Reply: Figure 4 was the map of Figure 3 when Figure 4 was looked upon along the u-axis. Former Figure 4 has been deleted, see the second reply to the first reviewer. 3: The reviewer wonders why $\gamma\_1$ is introduced in section '3.4 Flow patterns' Reply: the paragraph in which this was mentioned has been removed. Now $\gamma\_1$ is mentioned only where it is really used: in the paragraph explaining Figure 7, now at page 14.

Please also note the supplement to this comment:
https://www.wind-energ-sci-discuss.net/wes-2020-51/wes-2020-51-AC1-supplement.pdf

**Supplement:**

[revised manuscript text omitted]

---

## Author Response (AR1)

**Reply to the reviewers of WES-2020-51: On the velocity at wind turbine and propeller actuator discs, by G.A.M van Kuik**

The author thanks both reviewers for their positive as well as critical remarks. These have led to a real improvement of the paper, and to an extension with a paragraph on the radial velocity component. Below I reply to the comments of both reviewers, where after a the new version of the manuscript including indications of the changes in text and figures (please make the 'comment' tab active).

Reviewer RC1:

The reviewer asks attention for the radial velocity, with suggestions for additional literature.

> Reply: A subsection '4.3 The radial velocity' has been added, and the radial velocity component is now shown too in Fig. 5. The reviewer mentions a specific relation between the axial and radial velocity, derived in a manuscript-for-discussion by him and a co-author. This relation is mentioned, analysed and confirmed in section 4.3.

Furthermore the reviewer likes to see clearer the link with the author's previous work, specifically w.r.t. (previous) Figure 4.

> Reply : This figure has been deleted. The information considered essential for the line of thoughts is now included in a new version of Figure 3. This figure shows the flow regimes and the flow cases a-e. Moreover, the caption of this figure mentions the original figure in a previous publication.

Reviewer RC2:

The reviewer likes a clarification of the paragraph right after eq. (17), about the propeller flow without wake expansion of contraction. Especially the word 'flow state' was confusing.

> Reply: the words 'flow state' are not used any more. Instead the actuator disc flows that have been analysed in detail are called 'flow cases'. The paragraph right after eq. (17) has been adapted accordingly.

The reviewer finds it difficult to relate (former) Figure 4 to Figure 3.

> Reply: Figure 4 was the map of Figure 3 when Figure 4 was looked upon along the u-axis. Former Figure 4 has been deleted, see the second reply to the first reviewer.

The reviewer wonders why $\gamma_1$ is introduced in section '3.4 Flow patterns'

> Reply: the paragraph in which this was mentioned has been removed. Now $\gamma_1$ is mentioned only where it is really used: in the paragraph explaining Figure 7, now at page 14.

[revised manuscript text omitted]